# CoDA: Contrast-enhanced and Diversity-promoting Data Augmentation for Natural Language Understanding

**Yanru Qu**[1],[*] **Dinghan Shen**[2]**, Yelong Shen**[2]**, Sandra Sajeev**[2]**, Jiawei Han**[1]**, Weizhu Chen**[2]

[1]University of Illinois, Urbana-Champaign, [2]Microsoft Dynamics 365 AI
[1]`{yanruqu2,hanj}@illinois.edu`,
[2]`{dishen,yeshe,ssajeev,wzchen}@microsoft.com`

## Abstract

Data augmentation has been demonstrated as an effective strategy for improving model generalization and data efficiency. However, due to the discrete nature of natural language, designing label-preserving transformations for text data tends to be more challenging. In this paper, we propose a novel data augmentation framework dubbed *CoDA*, which synthesizes diverse and informative augmented examples by integrating multiple transformations organically. Moreover, a contrastive regularization objective is introduced to capture the *global* relationship among all the data samples. A momentum encoder along with a memory bank is further leveraged to better estimate the contrastive loss. To verify the effectiveness of the proposed framework, we apply CoDA to Transformer-based models on a wide range of natural language understanding tasks. On the GLUE benchmark, CoDA gives rise to an average improvement of $2.2\%$ while applied to the RoBERTa-large model. More importantly, it consistently exhibits stronger results relative to several competitive data augmentation and adversarial training baselines (including the low-resource settings). Extensive experiments show that the proposed contrastive objective can be flexibly combined with various data augmentation approaches to further boost their performance, highlighting the wide applicability of the CoDA framework.

## 1 Introduction

Data augmentation approaches have successfully improved large-scale neural-network-based models, (Laine & Aila, 2017; Xie et al., 2019; Berthelot et al., 2019; Sohn et al., 2020; He et al., 2020; Khosla et al., 2020; Chen et al., 2020b), however, the majority of existing research is geared towards computer vision tasks. The discrete nature of natural language makes it challenging to design effective label-preserving transformations for text sequences that can help improve model generalization (Hu et al., 2019; Xie et al., 2019). On the other hand, fine-tuning powerful, over-parameterized language models[1] proves to be difficult, especially when there is a limited amount of task-specific data available. It may result in representation collapse (Aghajanyan et al., 2020) or require special finetuning techniques (Sun et al., 2019; Hao et al., 2019). In this work, we aim to take a further step towards finding effective data augmentation strategies through systematic investigation.

In essence, data augmentation can be regarded as constructing neighborhoods around a training instance that preserve the ground-truth label. With such a characterization, adversarial training (Zhu et al., 2020; Jiang et al., 2020; Liu et al., 2020; Cheng et al., 2020) also performs label-preserving transformation in embedding space, and thus is considered as an alternative to data augmentation methods in this work. From this perspective, the goal of developing effective data augmentation strategies can be summarized as answering three fundamental questions:

*i*) What are some label-preserving transformations, that can be applied to text, to compose useful augmented samples?

---

[*]Work was done during an internship at Microsoft Dynamics 365 AI.

[1]To name a few, BERT (Devlin et al., 2019): 340M parameters, T5 (Raffel et al., 2019): 11B parameters, GPT-3 (Brown et al., 2020): 175B parameters.

*ii*) Are these transformations complementary in nature, and can we find some strategies to consolidate them for producing more diverse augmented examples?

*iii*) How can we incorporate the obtained augmented samples into the training process in an effective and principled manner?

Previous efforts in augmenting text data were mainly focused on answering the first question (Yu et al., 2018; Xie et al., 2019; Kumar et al., 2019; Wei & Zou, 2019; Chen et al., 2020a; Shen et al., 2020). Regarding the second question, different label-preserving transformations have been proposed, but it remains unclear how to integrate them organically. In addition, it has been shown that the diversity of augmented samples plays a vital role in their effectiveness (Xie et al., 2019; Gontijo-Lopes et al., 2020). In the case of image data, several strategies that combine different augmentation methods have been proposed, such as applying multiple transformations sequentially (Cubuk et al., 2018; 2020; Hendrycks et al., 2020), learning data augmentation policies (Cubuk et al., 2018), randomly sampling operations for each data point (Cubuk et al., 2020). However, these methods cannot be naively applied to text data, since the semantic meanings of a sentence are much more sensitive to local perturbations (relative to an image).

As for the third question, consistency training is typically employed to utilize the augmented samples (Laine & Aila, 2017; Hendrycks et al., 2020; Xie et al., 2019; Sohn et al., 2020; Miyato et al., 2018). This method encourages the model predictions to be invariant to certain label-preserving transformations. However, existing approaches only examine a pair of original and augmented samples in isolation, without considering other examples in the entire training set. As a result, the representation of an augmented sample may be closer to those of other training instances, rather than the one it is derived from. Based on this observation, we advocate that, in addition to consistency training, a training objective that can *globally* capture the intrinsic relationship within the entire set of original and augmented training instances can help leverage augmented examples more effectively.

In this paper, we introduce a novel **Co**ntrast-enhanced and **D**iversity-promoting Data **A**ugmentation (CoDA) framework for natural language understanding. To improve the diversity of augmented samples, we extensively explore different combinations of isolated label-preserving transformations in an unified approach. We find that *stacking* distinct label-preserving transformations produces particularly informative samples. Specifically, the most diverse and high-quality augmented samples are obtained by stacking an adversarial training module over the back-translation transformation. Besides the consistency-regularized loss for repelling the model to behave consistently within local neighborhoods, we propose a contrastive learning objective to capture the *global* relationship among the data points in the representation space. We evaluate CoDA on the GLUE benchmark (with RoBERTa (Liu et al., 2019) as the testbed), and CoDA consistently improves the generalization ability of resulting models and gives rise to significant gains relative to the standard fine-tuning procedure. Moreover, our method also outperforms various single data augmentation operations, combination schemes, and other strong baselines. Additional experiments in the low-resource settings and ablation studies further demonstrate the effectiveness of this framework.

## 2 METHOD

In this section, we focus our discussion on the natural language understanding (NLU) tasks, and particularly, under a text classification scenario. However, the proposed data augmentation framework can be readily extended to other NLP tasks as well.

### 2.1 BACKGROUND: DATA AUGMENTATION AND ADVERSARIAL TRAINING

**Data Augmentation** Let $\mathcal{D} = \{\boldsymbol{x}_i, y_i\}_{i=1...N}$ denote the training dataset, where the input example $\boldsymbol{x}_i$ is a sequence of tokens, and $y_i$ is the corresponding label. To improve model's robustness and generalization ability, several data augmentation techniques (*e.g.*, back-translation (Sennrich et al., 2016; Edunov et al., 2018; Xie et al., 2019), mixup (Guo et al., 2019), c-BERT (Wu et al., 2019)) have been proposed. Concretely, label-preserving transformations are performed (on the original training sequences) to synthesize a collection of augmented samples, denoted by $\mathcal{D}' = \{\boldsymbol{x}_i', y_i'\}_{i=1...N}$ . Thus, a model can learn from both the training set $\mathcal{D}$ and the augmented set $\mathcal{D}'$, with $p_\theta(\cdot)$ the predicted output distribution of the model parameterized by $\theta$:

$$\theta^* = \arg\min_\theta \sum_{(\boldsymbol{x}_i, y_i) \in \mathcal{D}} \mathcal{L}\big(p_\theta(\boldsymbol{x}_i), y_i\big) + \sum_{(\boldsymbol{x}_i', y_i') \in \mathcal{D}'} \mathcal{L}\big(p_\theta(\boldsymbol{x}_i'), y_i'\big) \tag{1}$$

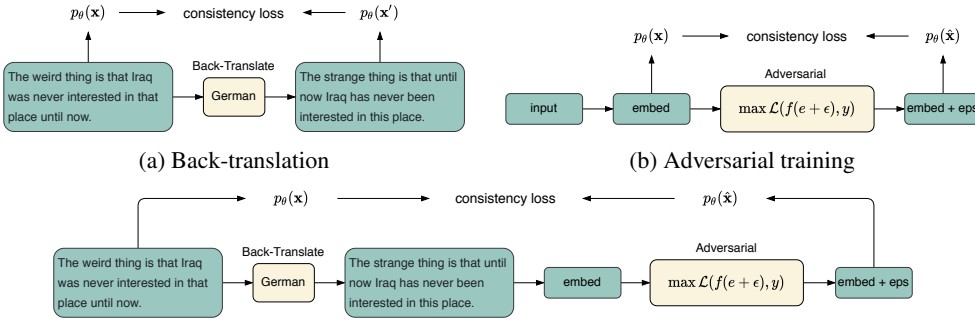

(a) Back-translation

(b) Adversarial training

(c) Stacking of back-translation and adversarial training

Figure 1: Illustration of data augmentation combined with adversarial training.

Several recent research efforts were focused on encouraging model predictions to be invariant to stochastic or domain-specific data transformations (Xie et al., 2019; Laine & Aila, 2017; Tarvainen & Valpola, 2017; Sohn et al., 2020; Miyato et al., 2018; Jiang et al., 2020; Hendrycks et al., 2020). Take back-translation as example: $\boldsymbol{x}_i' = \text{BackTrans}(\boldsymbol{x}_i)$, then $\boldsymbol{x}_i'$ is a paraphrase of $\boldsymbol{x}_i$. The model can be regularized to have consistent predictions for $(\boldsymbol{x}_i, \boldsymbol{x}_i')$, by minimizing the distribution discrepancy $\mathcal{R}_{\text{CS}}(p_\theta(\boldsymbol{x}_i), p_\theta(\boldsymbol{x}_i'))$, which typically adopts KL divergence (see Fig. 1a).

**Adversarial Training** In another line, adversarial training methods are applied to text data (Zhu et al., 2020; Jiang et al., 2020; Cheng et al., 2020; Aghajanyan et al., 2020) for improving model's robustness. Compared with data augmentation techniques, adversarial training requires no domain knowledge to generate additional training examples. Instead, it relies on the model itself to produce adversarial examples which the model are most likely to make incorrect predictions. Similar to data augmentation, adversarial training also typically utilizes the cross-entropy and consistency-based objectives for training. As the two most popular adversarial-training-based algorithms, the adversarial loss (Goodfellow et al., 2015) (Eqn. 2) and virtual adversarial loss (Miyato et al., 2018) (Eqn. 3) can be expressed as follows (see Fig. 1b):

$$\mathcal{R}_{\text{AT}}(\boldsymbol{x}_i, \tilde{\boldsymbol{x}}_i, y_i) = \mathcal{L}\big(p_\theta(\tilde{\boldsymbol{x}}_i), y_i\big), s.t., \|\tilde{\boldsymbol{x}}_i - \boldsymbol{x}_i\| \le \epsilon \,, \tag{2}$$

$$\mathcal{R}_{\text{VAT}}(\boldsymbol{x}_i, \tilde{\boldsymbol{x}}_i) = \mathcal{R}_{\text{CS}}\big(p_\theta(\tilde{\boldsymbol{x}}_i), p_\theta(\boldsymbol{x}_i)\big), s.t., \|\tilde{\boldsymbol{x}}_i - \boldsymbol{x}_i\| \le \epsilon \,. \tag{3}$$

Generally, there is no closed-form to obtain the exact adversarial example $\hat{\boldsymbol{x}}_i$ in either Eqn. 2 or 3. However, it usually can be approximated by a low-order approximation of the objective function with respect to $\boldsymbol{x}_i$. For example, the adversarial example in Eqn. 2 can be approximated by:

$$\hat{\boldsymbol{x}}_i \approx \boldsymbol{x}_i + \epsilon \frac{\boldsymbol{g}}{\|\boldsymbol{g}\|_2}, \text{where } \boldsymbol{g} = \nabla_{\boldsymbol{x}_i} \mathcal{L}\big(p_\theta(\boldsymbol{x}_i), y_i\big) \,. \tag{4}$$

## 2.2 DIVERSITY-PROMOTING CONSISTENCY TRAINING

As discussed in the previous section, data augmentation and adversarial training share the same intuition of producing neighbors around the original training instances. Moreover, both approaches share very similar training objectives. Therefore, it is natural to ask the following question: are different data augmentation methods and adversarial training equal in nature? Otherwise, are they complementary to each other, and thus can be consolidated together to further improve the model's generalization ability? Notably, it has been shown, in the CV domain, that combining different data augmentation operations could lead to more diverse augmented examples (Cubuk et al., 2018; 2020; Hendrycks et al., 2020). However, this is especially challenging for natural language, given that the semantics of a sentence can be entirely altered by slight perturbations.

To answer the above question, we propose several distinct strategies to combine different data transformations, with the hope to produce more diverse and informative augmented examples. Specifically, we consider 5 different types of label-preserving transformations: *back-translation* (Sennrich et al., 2016; Edunov et al., 2018; Xie et al., 2019), *c-BERT* word replacement (Wu et al., 2019), *mixup* (Guo et al., 2019; Chen et al., 2020a), *cutoff* (Shen et al., 2020), and *adversarial training* (Zhu et al., 2020; Jiang et al., 2020). The 3 combination strategies are schematically illustrated

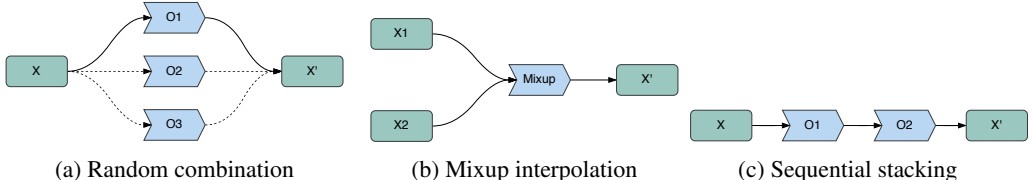

(a) Random combination        (b) Mixup interpolation        (c) Sequential stacking

Figure 2: Illustration of different strategies to combine various label-preserving transformations.

in Figure 2. For random combination, a particular label-preserving transformation is randomly selected, among all the augmentation operations available, for each mini-batch. As to the mixup interpolation, given two samples $\boldsymbol{x}_i$ and $\boldsymbol{x}_j$ drawn in a mini-batch, linear interpolation is performed between their input embedding matrices $\boldsymbol{e}_i$ and $\boldsymbol{e}_j$ (Zhang et al., 2017): $\boldsymbol{e}'_i = a\boldsymbol{e}_i + (1-a)\boldsymbol{e}_j$, where $a$ is the interpolation parameter, usually drawn from a *Beta* distribution.

Moreover, we consider stacking different label-preserving transformations in a sequential manner (see Figure 2c). It is worth noting that due to the discrete nature of text data, some stacking orders are infeasible. For example, it is not reasonable to provide an adversarially-perturbed embedding sequence to the back-translation module. Without loss of generality, we choose the combination where adversarial training is stacked over back-translation to demonstrate the sequential stacking operation (see Fig. 1c). Formally, given a training example $(\boldsymbol{x}_i, y_i)$, the consistency training objective for such a stacking operation can be written as:

$$\boldsymbol{x}'_i = \text{BackTrans}(\boldsymbol{x}_i), \ \hat{\boldsymbol{x}}_i \approx \text{argmax}_{\tilde{\boldsymbol{x}}_i} \mathcal{R}_{\text{AT}}(\boldsymbol{x}'_i, \tilde{\boldsymbol{x}}_i, y_i) , \tag{5}$$

$$\mathcal{L}_{\text{consistency}}(\boldsymbol{x}_i, \hat{\boldsymbol{x}}_i, y_i) = \mathcal{L}\big(p_\theta(\boldsymbol{x}_i), y_i\big) + \alpha\mathcal{L}(p_\theta(\hat{\boldsymbol{x}}_i), y_i) + \beta\mathcal{R}_{\text{CS}}(p_\theta(\boldsymbol{x}_i), p_\theta(\hat{\boldsymbol{x}}_i)) , \tag{6}$$

where the first term corresponds to the cross-entropy loss, the second term is the adversarial loss, $\mathcal{R}_{\text{CS}}$ denotes the consistency loss term between $(\boldsymbol{x}_i, \hat{\boldsymbol{x}}_i)$. Note that $\hat{\boldsymbol{x}}_i$ is obtained through two different label-preserving transformations applied to $\boldsymbol{x}$, and thus deviates farther from $\boldsymbol{x}$ and should be more *diverse* than $\boldsymbol{x}'_i$. Inspired by (Bachman et al., 2014; Zheng et al., 2016; Kannan et al., 2018; Hendrycks et al., 2020), we employ the Jensen-Shannon divergence for $\mathcal{R}_{\text{CS}}$, since it is upper bounded and tends to be more stable and consistent relative to the KL divergence:

$$\mathcal{R}_{\text{CS}}(p_\theta(\boldsymbol{x}_i), p_\theta(\hat{\boldsymbol{x}}_i)) = \frac{1}{2}\big(\text{KL}(p_\theta(\boldsymbol{x}_i)\|M) + \text{KL}(p_\theta(\hat{\boldsymbol{x}}_i))\|M)\big) , \tag{7}$$

where $M = (p_\theta(\boldsymbol{x}_i) + p_\theta(\hat{\boldsymbol{x}}_i))/2$. Later we simply use $\boldsymbol{x}'_i$ to represent the transformed example.

## 2.3 CONTRASTIVE REGULARIZATION

Consistency loss only provides local regularization, *i.e.*, $\boldsymbol{x}_i$ and $\boldsymbol{x}'_i$ should have close predictions. However, the relative positions between $\boldsymbol{x}'_i$ and other training instances $\boldsymbol{x}_j$ ($j \neq i$) have not been examined. In this regard, we propose to leverage a contrastive learning objective to better utilize the augmented examples. Specifically, we assume that the model should encourage an augmented sample $\boldsymbol{x}'_i$ to be closer, in the representation space, to its original sample $\boldsymbol{x}_i$, relative to other data points $\boldsymbol{x}_j$ ($j \neq i$) in the training set. This is a reasonable assumption since intuitively, the model should be robust enough to successfully determine from which original data an augmented sample is produced.

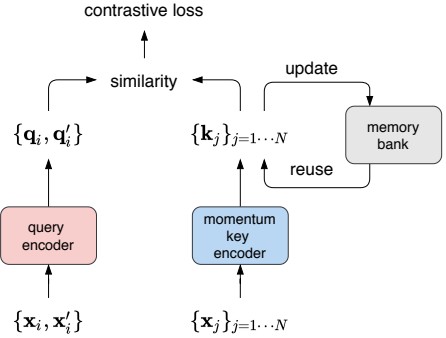

Figure 3: Illustration of the contrastive learning module.

The contrastive learning module is illustrated in Fig. 3. As demonstrated by prior efforts on contrastive learning, adopting a large batch size is especially vital for its effectiveness (Chen et al., 2020b; Khosla et al., 2020). Therefore, we introduce a memory bank that stores the history embeddings, thus enabling much larger number of negative samples. Moreover, to avoid the encoder from changing too rapidly (which may result in inconsistency embeddings), a momentum encoder module is incorporated into our algorithm. Concretely, let $f_\theta(.)$ and $f_{\bar{\theta}}(.)$ denote the transformation

parameterized by the query encoder and key encoder, respectively. Note that $\theta$ and $\bar{\theta}$ represent their parameters. The momentum model parameters $\bar{\theta}$ are not learned by gradients. Instead, they are updated through the momentum rule: $\bar{\theta} \leftarrow \gamma\bar{\theta} + (1 - \gamma)\theta$ at each training step. We omit the details here and refer the interested readers to the work by (He et al., 2020) for further explanation. Given a sample $\boldsymbol{x}_i$ and its augmented example $\boldsymbol{x}_i'$, the query and key can be obtained as follows:

$$\boldsymbol{q}_i = f_\theta(\boldsymbol{x}_i), \quad \boldsymbol{q}_i' = f_\theta(\boldsymbol{x}_i'), \quad \boldsymbol{k}_i = f_{\bar{\theta}}(\boldsymbol{x}_i) . \tag{8}$$

Thus, the contrastive training objective can be written as:

$$\mathcal{R}_{\text{contrast}}(\boldsymbol{x}_i, \boldsymbol{x}_i', \mathcal{M}) = \mathcal{R}_{\text{CT}}(\boldsymbol{q}_i, \boldsymbol{k}_i, \mathcal{M}) + \mathcal{R}_{\text{CT}}(\boldsymbol{q}_i', \boldsymbol{k}_i, \mathcal{M}), \tag{9}$$

$$\mathcal{R}_{\text{CT}}(\boldsymbol{q}_i, \boldsymbol{k}_i, \mathcal{M}) = -\log \frac{\exp(\text{sim}(\boldsymbol{q}_i, \boldsymbol{k}_i)/\tau)}{\sum_{\boldsymbol{k}_j \in \mathcal{M} \bigcup \{\boldsymbol{k}_i\}} \exp(\text{sim}(\boldsymbol{q}_i, \boldsymbol{k}_j)/\tau)}, \tag{10}$$

where $\tau$ is the temperature, and $\mathcal{M}$ is the memory bank in which the history keys are stored. Cosine similarity is chosen for $\text{sim}(\cdot)$. Note that $\mathcal{R}_{\text{CT}}(\boldsymbol{q}_i', \boldsymbol{k}_i, \mathcal{M})$ is similarly defined as $\mathcal{R}_{\text{CT}}(\boldsymbol{q}_i, \boldsymbol{k}_i, \mathcal{M})$ (with $\boldsymbol{q}_i$ replaced by $\boldsymbol{q}_i'$ in Eqn. 10). In Eqn. 9, the first term corresponds to the contrastive loss calculated on the original examples (self-contrastive loss), while the second term is computed on the augmented sample (augment-contrastive loss). Under such a framework, the pair of original and augmented samples are encouraged to stay closer in the learned embedding space, relative to all other training instances. As a result, the model is regularized *globally* through considering the embeddings of all the training examples available.

By integrating both the consistency training objective and the contrastive regularization, the overall training objective for the CoDA framework can be expressed as:

$$\theta^* = \text{argmin}_\theta \sum_{(\boldsymbol{x}_i, y_i) \in \mathcal{D}} \mathcal{L}_{\text{consistency}}(\boldsymbol{x}_i, \boldsymbol{x}_i', y_i) + \lambda \mathcal{R}_{\text{contrast}}(\boldsymbol{x}_i, \boldsymbol{x}_i', \mathcal{M}) . \tag{11}$$

where $\lambda$ is a hyperparameter to be chosen. It is worth noting that the final objective has taken both the *local* (consistency loss) and *global* (contrastive loss) information introduced by the augmented examples into consideration.

## 3 EXPERIMENTS

To verify the effectiveness of *CoDA*, We evaluate it on the widely-adopted GLUE benchmark (Wang et al., 2018), which consists of multiple natural language understanding (NLU) tasks. The details of these datasets can be found in Appendix B. RoBERTa (Liu et al., 2019) is employed as the testbed for our experiments. However, the proposed approach can be flexibly integrated with other models as well. We provide more implementation details in Appendix C. Our code will be released to encourage future research.

In this section, we first present our exploration of several different strategies to consolidate various data transformations (Sec 3.1). Next, we conduct extensive experiments to carefully select the contrastive objective for NLU problems in Sec 3.2. Based upon these settings, we further evaluate CoDA on the GLUE benchmark and compare it with a set of competitive baselines in Sec 3.3. Additional experiments in the low-resource settings and qualitative analysis (Sec 3.4) are further conducted to gain a deep understanding of the proposed framework.

### 3.1 COMBINING LABEL-PRESERVING TRANSFORMATIONS

We start by implementing and comparing several data augmentation baselines. As described in the previous section, we explore 5 different approaches: *back-translation*, *c-BERT* word replacement, *Mixup*, *Cutoff* and *adversarial training*. More details can be found in Appendix A. The standard cross-entropy loss, along with the consistency regularization term (Eq. 6) is utilized for all methods to ensure a fair comparison. We employ the MNLI dataset and RoBERTa-base model for the comparison experiments with the results shown in Table 1.

All these methods have achieved improvements over the RoBERTa-base model, demonstrating the effectiveness of leveraging label-preserving transformations for NLU. Moreover, back-translation, cutoff and adversarial training exhibit stronger empirical results relative to mixup and c-BERT.

To improve the diversity of augmented examples, we explore several strategies to combine multiple transformations: *i*) random combination, *ii*) mixup interpolation, and *iii*) sequential stacking, as

shown in Fig. 2. In Table 1, the score of naive random combination lies between single transformations. This may be attributed to the fact that different label-preserving transformations regularize the model in distinct ways, and thus the model may not be able to leverage different regularization terms simultaneously.

Besides, among all the other combination strategies, we observe that gains can be obtained by integrating back-translation and adversarial training together. Concretely, mixing back-translation and adversarial training samples (in the input embedding space) slightly improve the accuracy from $88.5$ to $88.6$. More importantly, the result is further improved to $88.8$ with these two transformations stacking together[2] (see Sec 2.2). With significance test, we find *stack (back, adv)* performs consistently better than other combinations (t-test of 10 runs, *p*-values $<$ 0.02). This observation indicates that the *stacking* operation, especially in the case of back-translation and adversarial training, can produce more diverse augment examples.

| Method | MNLI-m (Acc) | MMD |
|---|---|---|
| RoBERTa-base | 87.6 | - |
| *Single Transformation* | | |
| + back-translation | 88.5 | 0.63 |
| + c-BERT | 88.0 | 0.01 |
| + cutoff | 88.4 | 0.02 |
| + mixup (ori, ori) | 88.2 | 0.06 |
| + adversarial | 88.5 | 0.65 |
| *Multiple Transformations* | | |
| + random (back, cut, adv) | 88.4 | - |
| + mix (ori, back) | 88.4 | 0.11 |
| + mix (back, adv) | 88.6 | 0.81 |
| + stack (back, cut) | 88.5 | 0.62 |
| + stack (back, adv) | **88.8** | 1.14 |
| + stack (back, cut, adv) | 88.5 | 1.14 |
| + stack (back, adv, cut) | 88.4 | 1.14 |

Table 1: Comparison of different transformations on the MNLI-m development set. *Abbr:* original training instances (ori), back-translation (back), cutoff (cut), mixup (mix), adversarial (adv).

Intuitively, the augmented sample, with two sequential transformations, deviates more from the corresponding training data, and thus tends to be more effective at improving the model's generalization ability. To verify this hypothesis, we further calculate the MMD (Gretton et al., 2012) between augmented samples and the original training instances. It can be observed that *stack (back, adv)*, *stack (back, cut, adv)* and *stack (back, adv, cut)* have all produced examples the farthest from the original training instances (see Table 1). However, we conjecture that the latter two may have altered the semantic meanings too much, thus leading to inferior results. In this regard, *stack (back, adv)* is employed as the data transformation module for all the experiments below.

## 3.2 CONTRASTIVE REGULARIZATION DESIGN

In this section, we aim to incorporate the *global* information among the entire set of original and augmented samples *via* a contrastive regularization. First, we explore a few hyperparameters for the proposed contrastive objective. Since both the memory bank and the momentum encoder are vital components, we study the impacts of different hyperparameter values on both the temperature and the momentum. As shown in Fig. 4a, a temperature of 1.0 combined with the momentum of 0.99 can achieve the best empirical result. We then examine the size effect of the memory bank, and observe a larger memory bank size leads to a better capture of the global information and results in higher performance boost[3] (see Fig. 4b).

After carefully choosing the best setting based on the above experiments, we apply the contrastive learning objective to several GLUE datasets. We also implement several prior works on contrastive learning to compare, including the MoCo loss (He et al., 2020) and the supervised contrastive (SupCon) loss (Khosla et al., 2020), all implemented with memory banks. Note that we remove the consistency regularization for this experiment to better examine the effect of the contrastive regularization term (*i.e.*, $\alpha = \beta = 0$, $\lambda \neq 0$). As presented in Table 2, our contrastive objective consistently exhibits the largest performance improvement. This observation demonstrates for NLU, our data transformation module can be effectively equipped with the contrastive regularization.

---

[2]In practice, we can use Machine Translation (MT) models trained on large parallel corpus (*e.g.*, English-French, English-German) to back-translate the input sentence. Since back-translation requires decoding, it can be performed offline. If the input contains multiple sentences, we split it into sentences, perform back-translation, and ensemble those paraphrases back.

[3]We set the default memory bank size as 65536. As to smaller datasets, we choose the size no larger than the number of training data (*e.g.*, MRPC has 3.7k examples, and we set the memory size as 2048).

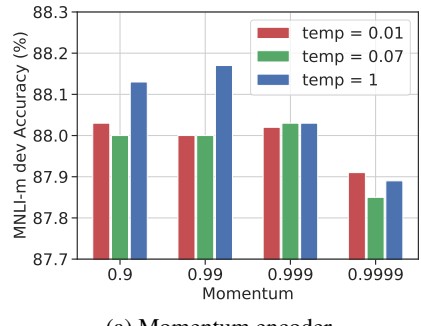
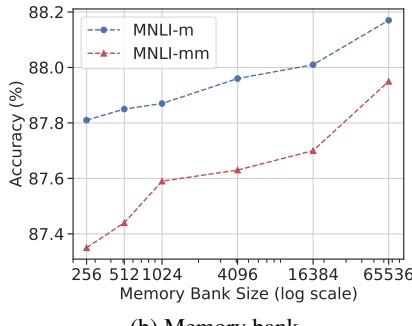

(a) Momentum encoder          (b) Memory bank

Figure 4: Hyperparameter exploration for the contrastive loss, evaluated on the MNLI-m development set. *Note:* All models use the RoBERTa-base model as the encoder.

| Method | MNLI-m (Acc) | QNLI (Acc) | SST-2 (Acc) | RTE (Acc) | MRPC (Acc) |
|---|---|---|---|---|---|
| RoBERTa-base | 87.6 | 92.8 | 94.8 | 78.7 | 90.2 |
| + MoCo (He et al., 2020) | **88.2** | 93.3 | 95.1 | 80.8 | 90.9 |
| + SupCon (Khosla et al., 2020) | 88.1 | 93.2 | 95.2 | 80.5 | 90.2 |
| + Contrastive (ours) | 88.1 | **93.6** | **95.3** | **82.0** | **91.7** |

Table 2: Comparison among different contrastive objectives on the GLUE development set.

## 3.3 GLUE BENCHMARK EVALUATION

With both components within the CoDA algorithm being specifically tailored to the natural language understanding applications, we apply it to the RoBERTa-large model (Liu et al., 2019). Comparisons are made with several competitive data-augmentation-based and adversarial-training-based approaches on the GLUE benchmark. Specifically, we consider back-translation, cutoff (Shen et al., 2020), FreeLB (Zhu et al., 2020), SMART (Jiang et al., 2020), and R3F (Aghajanyan et al., 2020) as the baselines, where the last three all belong to adversarial training. The results are presented in Table 3. It is worth noting that back-translation is based on our implementation, where both the cross-entropy and consistency regularization terms are utilized.

| Method | MNLI-m/ mm (Acc) | QQP (Acc/F1) | QNLI (Acc) | SST-2 (Acc) | MRPC (Acc/F1) | CoLA (Mcc) | RTE (Acc) | STS-B (P/S) | Avg |
|---|---|---|---|---|---|---|---|---|---|
| RoBERTa-large | 90.2/- | 92.2/- | 94.7 | 96.4 | -/90.9 | 68 | 86.6 | 92.4/- | 88.9 |
| Back-Trans | 91.1/90.4 | 92/- | 95.3 | 97.1 | 90.9/93.5 | 69.4 | 91.7 | 92.8/92.6 | 90.4 |
| Cutoff | 91.1/- | 92.4/- | 95.3 | 96.9 | 91.4/93.8 | 71.5 | 91.0 | 92.8/- | 90.6 |
| FreeLB | 90.6/- | **92.6**/- | 95 | 96.7 | 91.4/- | 71.1 | 88.1 | 92.7/- | - |
| SMART | 91.1/**91.3** | 92.4/89.8 | **95.6** | 96.9 | 89.2/92.1 | 70.6 | 92 | 92.8/92.6 | 90.4 |
| R3F | 91.1/**91.3** | 92.4/**89.9** | 95.3 | 97.0 | 91.6/- | 71.2 | 88.5 | - | - |
| CoDA | **91.3**/90.8 | 92.5/**89.9** | 95.3 | **97.4** | **91.9/94** | **72.6** | **92.4** | **93/92.7** | **91.1** |

Table 3: Main results of single models on the GLUE development set. *Note:* The best result on each task is in **bold** and "-" denotes the missing results. The average score is calculated based on the same setting as RoBERTa.

We find that *CoDA* brings significant gains to the RoBERTa-large model, with the averaged score on the GLUE dev set improved from 88.9 to 91.1. More importantly, CoDA consistently outperforms these strong baselines (indicated by a higher averaged score), demonstrating that our algorithm can produce informative and high-quality augmented samples and leverage them effectively as well. Concretely, on datasets with relatively larger numbers of training instances ($> 100K$), *i.e.*, MNLI, QQP and QNLI, different approaches show similar gains over the RoBERTa-large model. However, on smaller tasks (SST-2, MRPC, CoLA, RTE, and STS-B), CoDA beats other data augmentation or adversarial-based methods by a wide margin. We attribute this observation to the fact that the synthetically produced examples are more helpful when the tasks-specific data is limited. Thus, when smaller datasets are employed for fine-tuning large-scale language models, the superiority of the proposed approach is manifested to a larger extent.

## 3.4 ADDITIONAL EXPERIMENTS AND ANALYSIS

**Low-resource Setting** To verify the advantages of CoDA when a smaller number of task-specific data is available, we further conduct a low-resource experiment with the MNLI

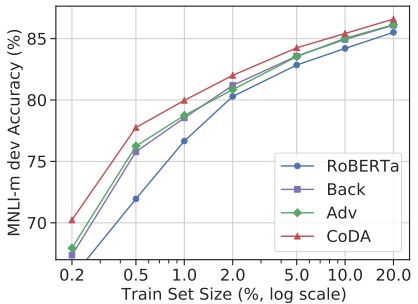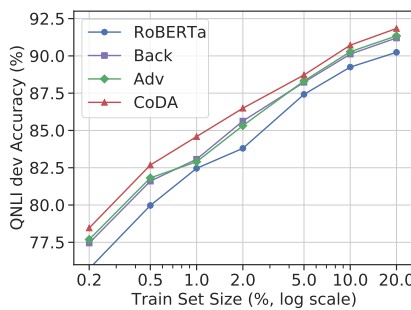

Figure 5: Low-resource setting experiments on the MNLI (left) and QNLI (right) dev sets.

and QNLI datasets. Concretely, different proportions of training data are sampled and utilized for training. We apply CoDA to RoBERTa-base and compare it with back-translation and adversarial training across various training set sizes. The corresponding results are presented in Fig. 5. We observe that back-translation and adversarial training exhibit similar performance across different proportions. More importantly, CoDA demonstrates stronger results consistently, further highlighting its effectiveness with limited training data.

**The Effectiveness of Contrastive Objective** To investigate the general applicability of the proposed contrastive regularization objective, we further apply it to different data augmentation methods. The RoBERTa-base model and QNLI dataset are leveraged for this set of experiments, and the results are shown in Fig. 6. We observe that the contrastive learning objective boosts the empirical performance of the resulting algorithm regardless of the data augmentation approaches it is applied to. This further validates our assumption that considering the *global* information among the embeddings of all examples is beneficial for leveraging augmented samples more effectively.

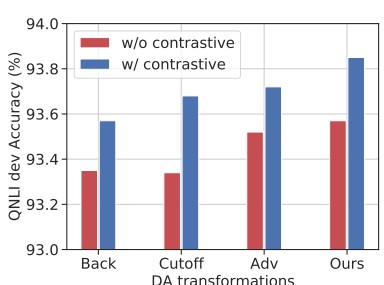

Figure 6: Evaluation of the proposed contrastive objective while applied to different data augmentation approaches.

## 4 RELATED WORK

**Data Augmentation in NLP** Different data augmentation approaches have been proposed for text data, such as back-translation (Sennrich et al., 2016; Edunov et al., 2018; Xie et al., 2019), c-BERT word replacement (Wu et al., 2019), mixup (Guo et al., 2019; Chen et al., 2020a), *Cutoff* (Shen et al., 2020). Broadly speaking, adversarial training (Zhu et al., 2020; Jiang et al., 2020) also synthesizes additional examples *via* perturbations at the word embedding layer. Although effective, how these data augmentation transformations may be combined together to obtain further improvement has been rarely explored. This could be attributed to the fact that a sentence's semantic meanings are quite sensitive to small perturbations. Consistency-regularized loss (Bachman et al., 2014; Rasmus et al., 2015; Laine & Aila, 2017; Tarvainen & Valpola, 2017) is typically employed as the training objective, which *ignores* the global information within the entire dataset.

**Contrastive Learning** Contrastive methods learn representations by contrasting positive and negative examples, which has demonstrated impressive empirical success in computer vision tasks (Hénaff et al., 2019; He et al., 2020). Under an unsupervised setting, Contrastive learning approaches learn representation by maximizing mutual information between local-global hidden representations (Hjelm et al., 2019; Oord et al., 2018; Hénaff et al., 2019). It can be also leveraged to learn invariant representations by encouraging consensus between augmented samples from the same input (Bachman et al., 2019; Tian et al., 2019). He et al. (2020); Wu et al. (2018) proposes to utilize a memory bank to enable a much larger number of negative samples, which is shown to benefit the transferability of learned representations as well (Khosla et al., 2020). Recently, contrastive learning was also employed to improve language model pre-training (Iter et al., 2020).

## 5 CONCLUSION

In this paper, we proposed CoDA, a Contrast-enhanced and Diversity promoting data Augmentation framework. Through extensive experiments, we found that stacking adversarial training over a back-translation module can give rise to more diverse and informative augmented samples. Besides,

we introduced a specially-designed contrastive loss to incorporate these examples for training in a principled manner. Experiments on the GLUE benchmark showed that CoDA consistently improves over several competitive data augmentation and adversarial training baselines. Moreover, it is observed that the proposed contrastive objective can be leveraged to improve other data augmentation approaches as well, highlighting the wide applicability of the CoDA framework.

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

## A    DATA AUGMENTATION DETAILS

We select the following representative data augmentation operations as basic building blocks of our data augmentation module. We denote $\boldsymbol{x}_i = [x_{i,1}, \ldots, x_{i,l}]$ as the input text sequence, and $\boldsymbol{e}_i = [\boldsymbol{e}_{i,1}, \ldots, \boldsymbol{e}_{i,l}]$ as corresponding embedding vectors.

- Back translation is widely applied in machine translation (MT) (Sennrich et al., 2016; Hoang et al., 2018; Edunov et al., 2018), and is introduced to text classification recently (Xie et al., 2019). Back-Trans uses 2 MT models to translate the input example to another pivot language, and then translate it back, $\boldsymbol{x}_i \rightarrow$ Pivot Language $\rightarrow \boldsymbol{x}'_i$.

- C-BERT Word Replacement (Wu et al., 2019) is a representative of the word replacement augmentation family. C-BERT pretrains a conditional BERT model to learn contextualized representation $P(\mathrm{x}_j | [x_{i,1} \ldots x_{i,j-1}[\mathrm{MASK}]x_{i,j+1} \ldots x_{i,l}], y_i)$ conditioning on classes. This method then randomly substitutes words of $\boldsymbol{x}$ to obtain $\boldsymbol{x}'$ $([x_{i,1} \ldots x'_{i,j} \ldots x_{i,l}])$[4].

- Cutoff (DeVries & Taylor, 2017) randomly drops units in a continuous span on the input, while Shen et al. (2020) adapts this method to text embeddings. For input embeddings $\boldsymbol{e}_i$, this method randomly set a continuous span of elements to 0s, $\boldsymbol{e}'_i = [\boldsymbol{e}_{i,1} \ldots \boldsymbol{e}_{i,j-1}, 0 \ldots 0, \boldsymbol{e}_{i,j+w} \ldots \boldsymbol{e}_{i,l}]$, where the window size $w \propto l$, and the start position $j \in [1, l-w]$ is randomly selected. For transformer encoders that involve position embeddings, we also set input mask as 0s at corresponding positions.

- Mixup (Zhang et al., 2017) interpolates two image as well as their labels. Guo et al. (2019) borrows this method to text. For 2 input embeddings $(\boldsymbol{e}_i, \boldsymbol{e}_j)$, mixup interpolates the embedding vectors $\boldsymbol{e}'_i = a\boldsymbol{e}_i + (1-a)\boldsymbol{e}_j$ where $a$ is sampled from a Beta distribution. Also, the labels are interpolated for the augmented sample $y'_i = ay_i + (1-a)y_j$.

- Adversarial training generates adversarial examples for input embeddings, simply, $\boldsymbol{e}'_i = \arg\max_{\|\boldsymbol{e}_i - \boldsymbol{e}'_i\| \leq 1} \mathcal{L}(f(\boldsymbol{e}'_i), y_i)$. We mainly follow the implementation of Zhu et al. (2020). Besides, when computing the adversarial example $\boldsymbol{e}'_i$, the dropout variables are recorded and reused later when encoding $\boldsymbol{e}'_i$.

Maximum mean discrepancy (MMD) (Gretton et al., 2012) is a widely used discrepancy measure for 2 distributions. We adopt the multi-kernel MMD implementation based on Shen et al. (2018)[5], to quantify the distance of data distributions before and after DA transformations.

## B    DATASET DETAILS

The datasets and statistics are summarized in Table 4.

| **Corpus** | Task | Sentence Pair | #Train | #Dev | #Test | #Class | Metrics |
|---|---|---|---|---|---|---|---|
| MNLI | NLI | ✓ | 393k | 20k | 20k | 3 | Accuracy |
| QQP | Paraphrase | ✓ | 364k | 40k | 391k | 2 | Accuracy/F1 |
| QNLI | QA/NLI | ✓ | 108k | 5.7k | 5.7k | 2 | Accuracy |
| SST | Sentiment | ✗ | 67k | 872 | 1.8k | 2 | Accuracy |
| MRPC | Paraphrase | ✓ | 3.7k | 408 | 1.7k | 2 | Accuracy/F1 |
| CoLA | Acceptability | ✗ | 8.5k | 1k | 1k | 2 | Matthews corr |
| RTE | NLI | ✓ | 2.5k | 276 | 3k | 2 | Accuracy |
| STS-B | Similarity | ✓ | 7k | 1.5k | 1.4k | - | Pearson/Spearman corr |

Table 4: GLUE benchmark summary.

---

[4]EDA (Wei & Zou, 2019) uses synonym replacement, another word replacement technique. We choose C-BERT for this family to take the advantages of contextual representation.

[5]https://github.com/RockySJ/WDGRL

## C    IMPLEMENTATION DETAILS

Our implementation is based on RoBERTa (Liu et al., 2019). We use ADAM (Kingma & Ba, 2014) as our optimizer. We follow the hyper-parameter study of RoBERTa and set as default the following parameters: batch size (32), learning rate (1e-5), epochs (5), warmup ratio (0.06), weight decay (0.1) and we keep other parameters unchanged with RoBERTa. For Back-Trans, we use the en-de single models trained on WMT19 and released in FairSeq. More specifically, we use beam search (beam size = 5) and keep only the top-1 hypothesis. We slightly tune Adversarial parameters on MNLI based on FreeLB and fix them on other datasets, since adversarial training is not our focus. For contrastive regularization, we implement based on MoCo. In GLUE evaluation, we mainly tune the weights of 3 regularization terms, $\alpha \in [0, 1], \beta \in [0, 3], \lambda \in [0, 0.03]$ (Eq. 6, 11). Besides, for smaller tasks (MRPC, CoLA, RTE, STS-B), we use the best performed MNLI model to initialize their parameters[6].

---

[6]RoBERTa:   https://github.com/huggingface/transformers.   FairSeq:   https://github.com/pytorch/fairseq. FreeLB: https://github.com/zhuchen03/FreeLB. MoCo: https://github.com/facebookresearch/moco.  We will release our model and code for further study.

