# OpenReview forum: "CoDA: Contrast-enhanced and Diversity-promoting Data Augmentation for Natural Language Understanding"
_ICLR.cc/2021/Conference — ICLR 2021 Poster_

### Official Review · AnonReviewer3 · 2020-10-28
**Comprehensive empirical evaluations and interesting ideas.**

**Rating:** 5
**Confidence:** 4

**Review:**

The paper proposes a novel data augmentation framework, which explores different combinations of isolated label-preserving transformations to improve the diversity of augmented samples.  The authors find that stacking distinct label-preserving transformations produces particularly informative samples.  The paper also introduces a contrastive learning objective to capture the global relationship among the data points in representation space.

In my opinion, the exploration of different combinations of isolated label-preserving transformations is the major contribution of this paper, which may inspire future works for data augmentation. However, the contrastive regularization object is a bit incremental, and I cannot see a big difference compared with Moco or SupCon.

Strength:

+ The idea of stacking distinct label-preserving transformations is intuitive.
+ The integration of the consistency training objective and the contrastive regularization objective is interesting.

Weakness:

- Lack of novelty, the contrastive regularization object is a bit incremental, and this object is very similar to MoCo or SupCon.
- The model has first to obtain the augmented samples, which is computation expensive for large-scale datasets and may hinder the practical application of the model. Moreover, the overall improvements are relatively small compared with R3F, and there is a lack of variance analysis.

Questions:

What is the computational complexity of CoDA?

Why using MMD distance in section 3.1?

Is stacking distinct label-preserving transformations the default setting for CoDA in your GLUE experiments? What if other strategies (mix, random) work better in datasets like QNLI, RTE. MRPC, and so on. Why not report results on those datasets?

What is the major difference between your contrastive regularization and MoCo or SupCon?

As the improvements are relatively small, could you please provide the test of statistical significance？

What if you stack cut first and then back? Does the order affect the performance?

---

> ### Author Response · Authors · 2020-11-18
> **Response to AnonReviewer3 (Part 1/2)**
>
> We would like to thank reviewer 3 for the valuable and thoughtful comments. Below we address the concerns/questions mentioned in the review:
>
> **Computational complexity:**
>
> Stack(back, adv) is chosen as the best performing combination strategy. Its computation overhead contains 3 parts:
>
> 1. Offline decoding of back translation. We use the wmt-2019 en-de single model of fairseq, where decoding MNLI (393k sentence pairs) takes 8 hours on a single 2080 GPU in total (with a beam size of 5). The data can be partitioned and fully paralleled on multiple GPUs, and the decoding is only performed once.
>
> 2. For the adversarial training module, we set the number of gradient ascent steps as 1. Thus, there is only one additional forward and backward pass for each mini-batch.
>
> 3. As to the contrastive regularization module, the original example is sent to the key encoder to obtain the corresponding embeddings as a separate step.  So it requires one additional forward pass.
>
> Summarizing, the computation complexity of CoDA is compared with other methods as below (note that S is the number of gradient ascent steps for adversarial training, and S >= 1):
>
> | Method               | # forward pass | # backward pass |
> |----------------------|----------------|-----------------|
> | Standard             |        1       |        1        |
> | FreeLB               |      1 + S     |      1 + S      |
> | SMART                |      1 + S     |      1 + S      |
> | R3F                  |        2       |        1        |
> | CoDA w/o contrastive |        2       |        2        |
> | CoDA w/ contrastive  |        3       |        2        |
>
> R3F, as a concurrent work to ours, shows promising efficiency and performance compared with other trust region methods. It would be interesting to replace our adversarial module with R3F, and we would expect better efficiency and results.
>
>
> **Why using MMD?**
>
> One of our main motivations is to improve the diversity of augmented samples by combining different data augmentation operations. To measure the diversity, we employ the MMD scores between CLS embeddings of the original and the augmented examples, where a higher value implies a higher distributional discrepancy. We choose MMD since it is easier to compute from data samples directly, unlike KLD or other measures which require estimating the density of CLS embeddings.
>
>
> **Is stacking the default setting for CoDA in GLUE experiments?**
>
> Yes. Stack(back, adv) is used as the default setting for all the experiments in Table 3.
>
>
> **What if other strategies (mix, random) work better in datasets? Why not report results on those datasets?**
>
> Thanks for the comments. We run the experiments with different combination strategies for other datasets (QNLI, MRPC, and RTE), and the results are as following:
>
> | Method                  | QNLI (acc) | MRPC (f1) | RTE (acc) |
> |-------------------------|------------|-----------|-----------|
> | Roberta                 |    94.7    |    90.9   |    86.6   |
> | +mix(ori, ori)          |    95.2    |    92.5   |    89.9   |
> | +mix(ori, back)         |    95.1    |    92.7   |    90.3   |
> | +mix(back, adv)         |    94.8    |    93.1   |    89.5   |
> | +random(back, cut, adv) |    95.2    |     93    |    89.5   |
> | +stack(back, adv)       |    95.3    |    93.5   |    91.7   |
>
> The reason we used MNLI is because it is widely recognized as a representative task of GLUE, thanks to its large volume, high annotation quality and wide coverage of different genres. We can also add the combination experiments for the remaining datasets as supplemental results.
>
>
> **What is the major difference between your contrastive regularization and MoCo or SupCon?**
>
> In table 2, we compare our contrastive regularization (ours) with MoCo and SupCon on 5 tasks. The major difference lies in how the query and key vectors are constructed.
>
> In MoCo, the query vector is the original example encoded by the query encoder, and the only positive key vector is the original example encoded by the key encoder. The negative keys are other examples encoded by the key encoder (they are stored in the memory bank).
> In SupCon, we replace its large batch implementation with memory bank and key encoder (the same as MoCo) due to limited GPU memory. SupCon has the same query vectors as MoCo, except that it adopts all key vectors obtained from the training examples with the same label as positive keys (with others treated as negatives).
>
> In our contrastive design, we include the proposed data augmentation module to validate whether our DA module is compatible with contrastive learning. The positive pairs are constructed by an original example and its corresponding augmented example. Other original and augmented samples are all treated as negatives. We tried to follow the SupCon setting and utilized all data with the same label to construct positive pairs, but it performs worse than our contrastive design.

---

> > ### Author Response · Authors · 2020-11-18
> > **Response to AnonReviewer3 (Part 2/2)**
> >
> > The results in table 2 indicate that:
> >
> > 1. The three contrastive regularizations generally improve Roberta.
> > 2. Our design achieves the best results on most datasets, demonstrating that the proposed data augmentation module can be effectively equipped with contrastive learning.
> >
> >
> > **Statistical significance and variance of results:**
> >
> > To validate that the proposed stack(back, adv) module is significantly better than other single operations or combination strategies, we perform t-test on MNLI through 10 runs with the same hyperparameters (see Section 3.1). Notably, the chosen setting performs significantly better relative to other data augmentation operations, with p-values < 0.02 in all cases. Considering the large size of the evaluation set for MNLI, this significance test should demonstrate that the gains are reliable.
> >
> > As to the variance of results, we run the experiments on MNLI and QNLI with both baseline and our approach for 5 times. The corresponding numbers, including the mean and variance, are shown as below:
> >
> > | Method         |      |      |      |      |      | avg   | std  |
> > |----------------|------|------|------|------|------|-------|------|
> > | MNLI (Roberta) | 90.2 | 90.3 | 90.3 | 90.5 | 90.4 | 90.34 | 0.1  |
> > | MNLI (CoDA)    | 91.3 | 91.2 | 91.2 | 91.2 | 91.3 | 91.24 | 0.05 |
> > | QNLI (Roberta) | 94.8 | 94.7 | 94.8 | 94.6 | 94.7 | 94.72 | 0.07 |
> > | QNLI (CoDA)    | 95.3 | 95.3 | 95.2 | 95.2 | 95.3 | 95.26 | 0.05 |
> >
> > It can be observed that CoDA consistently and reliably outperforms baseline according to the averaged number, even taking the results’ variance into consideration. We will run the same experiments for other datasets and add the results to a future version.
> >
> > **What if you stack cut first and then back? Does the order affect the performance?**
> >
> > We categorize basic DA operations into text-based (back translation, c-bert) and embedding-based (cutoff, mixup, adversarial) approaches. Since text-based methods require natural sentences as input, it is not reasonable to feed cutoff embeddings or cutoff sentences (missing words) to back translation or c-bert modules. Thus, in Section 3.1, we apply back translation before any embedding-based transformations, since embedding-based operations are more flexible and can take either text or embeddings as the input.The comparison between different orders we’ve tried is summarized in Table 1.

---

### Official Review · AnonReviewer4 · 2020-10-28
**A contrastive learning based framework for NLP Data augmenation**

**Rating:** 7
**Confidence:** 5

**Review:**

Paper proposes a contrastive learning-based approach to combine different data augmentation techniques for NLP tasks. While the widely used consistency loss focuses on a single example, the proposed contrastive objective allows capturing the relationships among all data samples which helps in producing diverse and informative examples. For experiments, the paper explores 5 data augmentation approaches with Roberta-large as the classification model. Empirical results on the standard GLUE benchmark leads to an impressive 2.2% average improvement. Authors also found that back-translation and adversarial training combination leads to better performance than other DA combinations.


Strengths:
1. The proposed framework can be applied with any text data augmentation methods. It's a solid work that will help the NLP community in developing better DA techniques. For example, [Kumar et al. 2020] shows that any pre-trained model can be used for data augmentation. I believe seq2seq model like T5, BART based augmentation combined with CoDA, will further push the state of the art for text DA.
2. Paper provides clear motivations and describes their methods, experiments in detail. Authors study DA in both low-resource and rich-resource setting. Ablation studies have been conducted to investigate gains from different components.
3. Authors plan to release their code which is good for reproducibility.

Weakness:
My understanding is that all numbers reported in the paper are from a single experiment. As a reader, I would like to see some variance with the results. Apart from this, I don't see any major issues with the paper.

Questions:
1. Since one of your goals is to improve the diversity of the augmented data, have you tried replacing more words in the c-bert model? By nature, c-bert is bound to replace max 15% of the tokens while maintaining the sentence length. Methods such as back-translation or seq2seq models do not have such restrictions.  Also, have you considered using a pre-trained seq2seq model for DA as in [Kumar et al. 2020]
2. Fig 5, back-translation, and adversarial training have similar performance. This result is intriguing.  Do you have some further insights into it?

Typos:
- Sec2.2. "the first term correspond" -> corresponds
- Sec 4, Contrastive Learning para, "ontrastive learning" -> "Contrastive learning"

References (additional DA citations):
1. Kumar, V., Choudhary, A., & Cho, E. (2020). Data Augmentation using Pre-trained Transformer Models. ArXiv, abs/2003.02245.

---

> ### Author Response · Authors · 2020-11-18
> **Response to AnonReviewer4**
>
> We would like to thank reviewer 4 for the valuable and thoughtful comments. Below we address the concerns/questions mentioned in the review:
>
>
>
> **Variance of results:**
>
> Great suggestion! We run the experiments on MNLI and QNLI with both baseline and our approach for 5 times. The corresponding numbers, including the mean and variance, are shown as below:
>
> | Method         |      |      |      |      |      | avg   | std  |
> |----------------|------|------|------|------|------|-------|------|
> | MNLI (Roberta) | 90.2 | 90.3 | 90.3 | 90.5 | 90.4 | 90.34 | 0.1  |
> | MNLI (CoDA)    | 91.3 | 91.2 | 91.2 | 91.2 | 91.3 | 91.24 | 0.05 |
> | QNLI (Roberta) | 94.8 | 94.7 | 94.8 | 94.6 | 94.7 | 94.72 | 0.07 |
> | QNLI (CoDA)    | 95.3 | 95.3 | 95.2 | 95.2 | 95.3 | 95.26 | 0.05 |
>
> It can be observed that CoDA consistently and reliably outperforms baseline according to the averaged number, even taking the results’ variance into consideration. We will run the same experiments for other datasets and add the results to a future version.
>
>
>
> **Question 1:**
>
> The ratio of c-bert is not tuned in our experiments, since 15% is suggested by the official c-bert code release. We agree that data augmentation methods like back-translation or seq2seq models do not have any length restrictions. As a result, they may be able to generate more diverse examples (relative to c-bert). We thank the reviewer for pointing out this interesting reference, and we would be happy to include it as a related work in the updated version.
>
>
>
> **Question 2:**
>
> It is an interesting observation that back translation and adversarial training demonstrate close performance gains. One potential explanation may be that the diversity of augmented samples with these two methods are very similar. Specifically, as shown in table 1, the two methods have very similar MMD scores (0.63 and 0.65), which measure the distribution divergence between the CLS embeddings of the original and the augmented examples.
>
>
>
> **Other Responses:**
>
> We will fix the typos and add the missing reference to a future version.

---

### Official Review · AnonReviewer1 · 2020-10-28
**Official Blind Review #1**

**Rating:** 6
**Confidence:** 4

**Review:**

Summary:

The augmentation of NLP samples is an important task with no clear "applicable to all" mechanism. This is in sharp contrast to computer vision where techniques like rotation, modification of hue, saturation as well as umpteen other techniques exist. This work tries to address the issue by proposing a technique that carefully amalgamates multiple previously known approaches to generate diverse label preserving examples. The experimental results on RoBERTa highlight the applicability and importance of this data augmentation approach on the downstream task of text classification (GLUE).

Strengths:

1. Empirical results. Performance better than previous approaches (although minor).
2. Paper Clarity
3. Each formulation is backed by a strong intuitive understanding.
4. Contrastive training (negative sampling) is one of the crucial contributions of this work. It seems to be making every previously known augmentation approach better.
Please feel free to highlight other major contributions.

Weaknesses (Minor):

1. Ad-hoc regularization parameter selection is necessary for getting performance gains. This makes it difficult to conclusively prove that this is an "applicable to all" data augmentation scheme.
2. It would have been better to see the performance gains on more difficult text-classification tasks (non-GLUE), or underperforming models (non-BERT based). Since the gains are not much. It becomes difficult to fathom if the gains are actually due to good objective function or a case of chance for choosing better examples.


Comments/Questions:

1. What is the augmentation size being used in the setup? I suspect the size plays an important role in such setups and this hasn't been discussed much in the paper. Also, please show the performance trends based on different augmentation sizes.

2. How do you measure the diversity (as mentioned in the paper title) in the generated samples?

3. Rather than using the ad-hoc approach for selecting which augmentation "stacking" scheme is helpful, it would have been better to compare/use an approach highlighted in "Learning to Compose Domain-Specific Transformations for Data Augmentation" [NeuRIPS 2017].


Correction:

1. Related Work: Contrastive learning - Under an unsupervised setting, ontrastive -> contrastive

Overall:

This work highlights the importance of incorporating contrastive training for data augmentation.

Please let me know if I have misunderstood something(s)

---

> ### Author Response · Authors · 2020-11-18
> **Response to AnonReviewer1**
>
> We would like to thank reviewer 1 for the valuable and thoughtful comments. Below we address the concerns/questions mentioned in the review:
>
>
>
>
> **Ad-hoc regularization parameters:**
>
> On the selection of hyperparameters, the default of the three regularization terms are set as (alpha=0.3, beta=1, lambda=0.01), which can generally achieve the reported results on most datasets. One exception here is MRPC, where we find that a larger lambda (i.e., 0.3) tends to perform even better. Thus, the results are not very sensitive to these hyperparameters in general.
>
>
>
>
> **More difficult settings:**
>
> Thanks for your great suggestion. We agree that evaluating our approach on more challenging tasks/datasets and non-BERT based models may demonstrate larger gains. Besides, it is worth noting that our method is tested in a challenging low-resource setting (see Section 3.4), i.e., we compare Roberta and Roberta+CoDA when they are both trained with a very small fraction of the original training samples. In the most difficult setting (0.2% of the training data is used, which means around 800 examples for MNLI and about 200 examples for QNLI), CoDA still outperforms Roberta by a wide margin. That said, we will consider adding other difficult settings to complement the evaluations later.
>
> Moreover, in Section 3.1, we perform t-test on MNLI through 10 runs of the same hyperparameters. Notably, the chosen setting performs significantly better relative to other data augmentation operations, with p-values < 0.02 in all cases. Considering the large size of the evaluation set for MNLI, this significance test should demonstrate that the gains are reliable.
>
>
>
>
> **Diversity measure:**
>
> Due to the discrete nature of natural language, it is quite challenging to measure its diversity.
> Since most of the augmentation transformations considered here involve perturbations in the input embedding space, token-based diversity metrics do not apply here. As a result, to measure the diversity, we calculate the Maximum Mean Discrepancy (MMD) between the CLS embeddings of the original and augmented examples. The corresponding results are shown in Table 1. We observe that stacking multiple operations together typically results in higher MMD scores, implying that the augmented samples are further from the original training data. Notably, stacking back-translation and adversarial training achieves the highest MMD score as well as the best results on GLUE. Therefore, we suppose that MMD is a reasonable measure regarding the diversity and effectiveness of augmented examples.
>
>
>
>
> **Other responses:**
>
> For the ratio between augmented samples and original training data, {1,2,3} are tried on back translation, cutoff, and mixup in our initial experiments. We did not try larger sizes due to GPU memory limits. It was found that the size of 1 already performed reasonably well and there was no much gain by adding more samples. Besides, considering that the top-1 hypothesis of back translation usually has the best quality, and there is only one adversarial example inferred from each input, we set the ratio as 1 for all other experiments.
>
> We thank the reviewer for pointing out the typo and the valuable reference. The reference shows another promising direction for developing even more effective data augmentation techniques, and we would like to leave this as future work. That said, we'll happily discuss the reference that you suggested in a future version.

---

> > ### Comment · AnonReviewer1 · 2020-11-21
> > **Follow Up**
> >
> > Thanks for the response. I just have some follow up questions.
> >
> > 1. Would you like to elaborate more on the choice of hyper-parameters. Why do you think might MRPC require a higher lambda than the other settings? Also, how would one measure the insensitivity to hyper-parameters? Is that instance-specific i.e., the outputs obtained using different hyper-params are the same (or lexically/spatially similar) OR corpus-specific i.e., the final accuracy score remains the same although the outputs for each instance are diverse?
> >
> > 2. Confusion regarding different ratios: I am not sure how the memory size of the GPU would affect trying different augmentation sizes. The overall data set size surely increases but the batch which one operates with, remains under control. I believe, in the end, the classification model is being trained using mini-batch;  unless you are loading all the data points into memory at once and performing full gradient descent. Please let me know in case I have misunderstood this point.
> >
> > 3. Can you comment on the choice of data points being used for augmentation. I am anticipating that since the performance gains are not really observable when you increase the ratio, there must be a smaller subset of points (that you can select for augmentation) that will help you reach the same scores. If so, is there a way to select those points? This is tied to the concern that the artificially induced low resource setting for training Roberta and Roberta + CODA might be just working with those points. Since this is different from having actual low-resource data, a badly (arguably) selected subset will result in poor performance.

---

> > > ### Author Response · Authors · 2020-11-24
> > > **Response to Follow Up**
> > >
> > > 1. Most of the hyper-parameters are detailed in Appendix C. More specifically, we tune alpha = {0, 0.3, 1}, beta = {0, 0.3, 1, 3}, lambda = {0, 0.01,0.03} in main experiments. The following table shows one of the parameter search results with stack(back, adv) (no contrastive regularization):
> > >
> > > | MNLI-m (acc) | alpah = 0 | alpha = 0.3 | alpha = 1 |
> > > |--------------|-----------|-------------|-----------|
> > > | beta = 0     |    90.2   |    90.89    |   91.15   |
> > > | beta = 0.3   |   90.86   |    90.98    |   91.03   |
> > > | beta = 1     |   90.88   |    91.25    |   91.19   |
> > > | beta = 3     |   90.92   |    91.15    |   91.05   |
> > >
> > > From the table we can see, final accuracy scores are stable in general.
> > >
> > > With “insensitive”, we refer to the fact that the final accuracy scores are insensitive to those hyper-parameters, and our performance gains are not by chance. In terms of the output diversity, we compare the MNLI-m dev set predictions from 2 checkpoints, i.e., (alpha = 0.3, beta = 1) and (alpha = 1, beta = 1). The results show that among all 9815 examples, only 252 predictions are flipped when using different hyperparameters.
> > >
> > >
> > > 2. Yes, we adopt mini-batch training. For each mini-batch, the same number of original examples (i.e., n examples) are included. With different ratios k, we augment each original example with k examples. Thus each mini-batch will contain (k+1)*n examples in total, which grows linearly with ratio k.
> > >
> > >
> > > 3. We generally augment all training data points. And in low resource experiments, we take the data points in the front for training.
> > >
> > > This is an interesting point. We agree that leveraging augmented samples for a small subset of data points may reach the same performance. It is a research question worth exploring, and we would like to leave it for future work.

---

### Comment · ~Jaehyung_Kim1 · 2021-03-13
**Open source code**

Hi, all,

Thank you for your great work. I am interested in the proposed method.

Would you please share your code that can reproduce the proposed algorithm?

Thank you very much!

---

> ### Comment · ~Yanru_Qu1 · 2022-03-27
> **Release source code soon**
>
> Sorry for the late response, since I was seriously ill and hospitalized for a long time.
>
> I will soon back to work and release the code.
>
> Thanks,
> Yanru

---

### Comment · ~Hai_Ye2 · 2021-06-01
**A little question**

Thanks for the interesting work.

Just a little question:

In the appendix, you mention for MRPC, CoLA, RTE, STS-B, you initialize the model with the one pre-trained on MNLI, is it commonly used by the baselines you compared in Table 3?  Such that the RoBERTa-baseline has done the same?

Thanks.

---

> ### Comment · ~Yanru_Qu1 · 2022-03-27
> **Initialize with MNLI pretrained model**
>
> Sorry for the late response, since I was seriously ill and hospitalized for a long time.
>
> For those small datasets mentioned in the paper, we initialize there roberta-large model with an MNLI pretrained one. For other larger datasets, we initialize with roberta-large.

---

### Decision · Program_Chairs · 2021-01-07
**Final Decision**

**Decision:**

Accept (Poster)

**Comment:**

This paper concerns data augmentation techniques for NLP. In particular, the authors introduce a general augmentation framework they call CoDA and demonstrate its utility on a few benchmark NLP tasks, reporting promising empirical results. The authors addressed some key concerns (e.g., regarding hyperparameters, reporting of variances) during the discussion period. The consensus, then, is that this work provides a useful and relatively general method for augmentation in NLP and the ICLR audience is likely to find this useful.